# Deprescribing interventions in older adults: An overview of systematic reviews

**Shiyun Chua[1☯¤], Adam Todd[2,3‡], Emily Reeve [4,5‡], Susan M. Smith[6‡], Julia Fox[7],
Zizi Elsisi[7], Stephen Hughes [8], Andrew Husband[2,3], Aili Langford [4], Niamh Merriman [6],
Jeffrey R. Harris[1], Beth Devine[7☯], Shelly L. Gray [7,9☯]\*, the Expert Panel[¶]**

**1** School of Public Health, University of Washington, Seattle, Washington, United States of America,
**2** Newcastle University, School of Pharmacy, Newcastle-upon-Tyne, United Kingdom, **3** NIHR Patient Safety
Research Collaborative, Newcastle-upon-Tyne, United Kingdom, **4** Centre for Medicine Use and Safety,
Monash Institute of Pharmaceutical Sciences, Faculty of Pharmacy and Pharmaceutical Sciences, Monash
University, Melbourne, VIC, Australia, **5** Quality Use of Medicines and Pharmacy Research Centre, Clinical
and Health Sciences, University of South Australia, Adelaide, Australia, **6** Discipline of Public Health and
Primary Care, School of Medicine, Trinity College Dublin, Dublin, Ireland, **7** School of Pharmacy, University of
Washington, Seattle, Washington, United States of America, **8** School of Pharmacy, University of Sydney,
Sydney, Australia, **9** Plein Center for Geriatric Pharmacy Research, Education and Outreach, School of
Pharmacy, University of Washington, Seattle, Washington, United States of America

☯ These authors contributed equally to this work.
¤ Current address: Tan Tock Seng Hospital, Singapore, Singapore
‡ AT, ER and SMS also contributed equally to the work.
¶ Membership of the Expert Panel is provided in the Acknowledgments.
\* slgray@uw.edu

Medical Sciences, ISLAMIC REPUBLIC OF IRAN

**Data Availability Statement:** All relevant data are
within the manuscript and its Supporting
Information files.

**Funding:** This study was supported by National
Institute on Aging (NIA: 1R24AG064025, MPI

## Abstract

### Objective

The growing deprescribing field is challenged by a lack of consensus around evidence and
knowledge gaps. The objective of this overview of systematic reviews was to summarize the
review evidence for deprescribing interventions in older adults.

### Methods

11 databases were searched from 1$^{st}$ January 2005 to 16$^{th}$ March 2023 to identify system-
atic reviews. We summarized and synthesized the results in two steps. Step 1 summarized
results reported by the included reviews (including meta-analyses). Step 2 involved a narra-
tive synthesis of review results by outcome. Outcomes included medication-related out-
comes (*e.g.*, medication reduction, medication appropriateness) or twelve *other* outcomes
(*e.g.*, mortality, adverse events). We summarized outcomes according to subgroups
(patient characteristics, intervention type and setting) when direct comparisons were avail-
able within the reviews. The quality of included reviews was assessed using A MeaSure-
ment Tool to Assess systematic Reviews 2 (AMSTAR 2).

### Results

We retrieved 3,228 unique citations and assessed 135 full-text articles for eligibility. Forty-
eight reviews (encompassing 17 meta-analyses) were included. Thirty-one of the 48 reviews
had a general deprescribing focus, 16 focused on specific medication classes or therapeutic

Steinman, Boyd; SG Co-investigator). www.nia.nih.gov. The views expressed are those of the author(s) and not necessarily those of the NIA. The funders had no role in considering the study design or in the collection, analysis, and interpretation of data, the writing of the report, or the decision to submit the article for publication. ER and AL was supported by an Australian National Health and Medical Research Council (NHMRC) Investigator Grant (APP1195460). NM was supported by Health Research Board Collaboration in Ireland for Clinical Effectiveness Reviews Award (HRB-CICER-2016-1871).

**Competing interests:** The authors have declared that no competing interests exist.

**Abbreviations:** ADR, Adverse drug reaction; ADWE, Adverse drug withdrawal events; AMSTAR 2, A MeaSurement Tool to Assess systematic Reviews 2; APA, American Psychological Association; CCA, Corrected Coverage Area; CINAHL, Cumulative Index to Nursing and Allied Health Literature; DARE, Database of Abstracts of Reviews of Effects; GRADE, Grading of Recommendations, Assessment, Development and Evaluations; HRQoL, Health Related Quality of Life; MAI, Medication Appropriateness Index; NHS EED, National Health Service Economic Evaluation Database; PICOS, Population, Intervention, Comparison, Outcomes, Study Design; PIM, Potentially inappropriate medication; PRISMA, Preferred Reporting Items for Systematic Reviews and Meta-Analyses; PROMs, Patient-reported outcome measures; STOPP, Screening Tool of Older Persons' Prescriptions.

categories and one included both. Twelve of 17 reviews meta-analyzed medication-related outcomes (33 outcomes: 25 favored the intervention, 7 found no difference, 1 favored the comparison). The narrative synthesis indicated that most interventions resulted in some evidence of medication reduction while for *other* outcomes we found primarily no evidence of an effect. Results were mixed for adverse events and few reviews reported adverse drug withdrawal events. Limited information was available for people with dementia, frailty and multimorbidity. All but one review scored low or critically low on quality assessment.

## Conclusion

Deprescribing interventions likely resulted in medication reduction but evidence on *other* outcomes, in particular relating to adverse events, or in vulnerable subgroups or settings was limited. Future research should focus on designing studies powered to examine harms, patient-reported outcomes, and effects on vulnerable subgroups.

## Systematic Review Registration

PROSPERO CRD42020178860.

## Introduction

More than 40% of older adults aged ≥75 years in developed countries are prescribed five or more medications on a regular basis [1]. Use of multiple medications, or polypharmacy, is associated with increased risk of poor outcomes [2–4]. These harms appear to be amplified in vulnerable subgroups [5], such as those with frailty [6, 7], or dementia [8]. Moreover, approximately 50% of older adults are estimated to receive at least one potentially inappropriate medication (PIM) [9]. For decades, research has focused on mitigating the harmful effects of polypharmacy by reducing the number of medications and discontinuing those where harms outweigh benefits. More recently, deprescribing has emerged as a systematic approach for improving the quality of medication use that is patient-centered and is aligned with the 4Ms of Age-Friendly care (mind, mobility, medications, and what matters most) [10]. Although definitions vary, deprescribing is the process of discontinuing, or reducing the dose of, medications that are no longer needed, or where risks outweigh benefits or are inconsistent with goals of care. Deprescribing is a process supervised by health care professionals with the goal of managing polypharmacy and improving health outcomes [9, 11].

Numerous studies and systematic reviews (referred to as reviews hereafter) have examined a variety of strategies to deprescribe in older adults, with varying levels of success [12, 13]. Despite this progress, rigorous evidence to guide deprescribing is limited, and future priorities for deprescribing research and practice remain unclear. Challenges contributing to the heterogeneity of evidence have included the lack of a consistent definition of deprescribing [14], challenges with outcome measurement, poor reporting of studies, and wide variation in study design [4]. Further, it is unclear to what extent vulnerable patients, including people with frailty and dementia, have been included in deprescribing trials.

The availability of reviews in deprescribing has increased in recent years. These reviews are diverse in the medication focus of deprescribing, included patient populations, and setting [13, 15]. Given this, a broad and comprehensive summary of deprescribing reviews would increase understanding of the areas of deprescribing with the highest potential to improve

patient outcomes to inform allocation of limited healthcare resources and research funding. An overview of reviews is an established way to examine a broader scope by synthesizing the breadth of research available, elucidating key findings, and identifying gaps and future research priorities, from a field with a rapid increase in the number of diverse reviews [16, 17]. Our objective was to summarize the review evidence for deprescribing interventions in older adults. We sought to synthesize review data according to medication focus of the intervention (i.e., specific medication class or therapeutic category versus general deprescribing) by outcomes of interest. In addition, we summarized review findings by pre-determined subgroups (patient characteristics, intervention type and setting) when direct comparisons were available within reviews.

## Material and methods

The protocol was registered with PROSPERO (CRD42020178860) and the methods were guided by the Preferred Reporting Items for Systematic Reviews and Meta-Analyses (PRISMA) guidelines [18] and the Cochrane Handbook chapter on overviews [16]. Amendments to the protocol can be found in S1 Table.

### Search strategy

We developed a search strategy in collaboration with an information specialist (e.g., medical librarian), and executed it in Medline, Embase, Cumulative Index to Nursing and Allied Health Literature (CINAHL) Complete, American Psychological Association (APA) PsycInfo, Scopus, Web of Science Core Collection, Cochrane Database of Systematic Reviews, Database of Abstracts of Reviews of Effects (DARE), Health Technology Assessment, National Health Service Economic Evaluation Database (NHS EED), and Epistemonikos. Databases were searched from January 1st, 2005, to March 16th, 2023 without language restrictions. Search terms were tailored to the specific host site, with a combination of free keywords and MeSH terms (S1 Appendix). Reference lists of relevant articles were hand-searched for potentially relevant reviews.

### Selection criteria

We developed a conceptual model to guide the design of the review (S1 Fig). The inclusion criteria were determined *a priori* by using the population, intervention, control, outcome, study design and setting (PICOS) framework (Table 1). Reviews were excluded if not available in English or did not contain at least two eligible primary studies (i.e., primary studies meeting the PICOS criteria).

Outcomes of interest were adapted from the core outcome set for polypharmacy research, because there is no specific core outcome set for deprescribing (Table 2) [23].

### Data extraction

The titles and abstracts of returned studies were screened independently for eligibility for inclusion by at least two reviewers (SC, JF, ZE, AL, NM, YH), with disagreements resolved by consensus with a third reviewer (SG). The same process was used for full text reviews.

Data from each review were independently extracted into Excel® (Redmond, WA) by two reviewers (SC, JF, ZE, AL, NM, YH) using a bespoke data-extraction form. Extracted data included review characteristics, intervention, focus of the intervention (specific medications or general deprescribing), outcomes (and their GRADE [24] assessment of quality of evidence across studies for the outcome where available), meta-analysis results, subgroup data, and authors' interpretation of results obtained from the discussion and conclusion. Information regarding PICOS was extracted from the methods section.

**Table 1. Population, intervention, comparison, outcomes, study design and setting.**

| P | Population | Studies wherein the mean age of participants was 60 years and older. Reviews were included if they did not focus on older adults, if we could identify primary studies where the mean age ≥60 and only these primary studies were considered eligible. Studies conducted in long-term care facilities were also included even if the age was not provided given most people in these settings meet this age criteria. |
|---|---|---|
| I | Intervention | Any intentional action or strategy designed to result in deprescribing. The following types of interventions were included:<br>• Interventions with a deprescribing focus:<br> ○ Specific medications (*e.g.*, single medication, medication class or therapeutic category).<br> ○ General deprescribing (*e.g.*, comprehensive medication reviews, interventions to reduce polypharmacy or PIMs).<br>• Interventions of investigator-initiated medication withdrawal. |
| C | Comparison | Reviews including comparisons with 'usual care' and/or medication continuation. |
| O | Outcomes | For inclusion, reviews needed to have a medication-related outcome. We had 12 outcomes of interest (Table 2) which were grouped into two categories: 1) Medication-related outcomes and 2) *Other* outcomes |
| S | Study design | Reviews including randomized trials, non-randomized trials, controlled before-after studies, interrupted time-series studies and repeated-measures studies [19]. Following the methods of previous overviews [20, 21], we excluded records that did not meet the ≥4 DARE criteria [22] for a systematic review.[a] |
| S | Setting | Any health care setting in any country |

PIM, potentially inappropriate medication; DARE, the Database of Abstracts of Reviews of Effects.

[a] DARE specifies at least four of the following criteria must be met: 1. Were inclusion/exclusion criteria reported?, 2. Was the search adequate?, 3. Were the included studies synthesized?, 4. Was the quality of the included studies assessed? and 5. Are sufficient details about the individual included studies presented?

**Table 2. Description of medication-related outcomes and *other* outcomes.**

| | Outcomes |
|---|---|
| Medication-related Outcomes | Medication reduction<br>• number of medications prescribed, pre- and post-intervention, dose reduction, discontinuation, potentially inappropriate medications (*e.g.*, STOPP criteria) |
| | Medication appropriateness (*e.g.*, MAI) |
| Other Outcomes | Surrogate biomarkers<br>• HbA1c, blood pressure |
| | Mortality |
| | Health-related quality of life (HRQoL)[a] |
| | Patient perception of treatment burden |
| | Cognition |
| | Falls |
| | Hospitalizations |
| | Costs<br>• hospitalization and/or emergency costs |
| | Adverse events<br>• adverse drug reactions (ADR)[b] and adverse drug withdrawal events ADWE) |
| | Other patient-reported outcome measures<br>• *e.g.*, activities of daily living, symptom scores for sleep, neuropsychiatric symptoms |

MAI, medication appropriateness index; STOPP, Screening Tool of Older Persons' Prescriptions.

[a] We used the term HRQoL as defined by the review authors.

[b] Studies may have used the term adverse drug reactions or adverse drug event interchangeably. We use the term adverse drug reactions to refer to both, despite these being different concepts.

### Assessment of methodologic quality and study overlap

Quality assessment was performed independently by 2 reviewers (YH, SC, ZE, SH) using A MeaSurement Tool to Assess systematic Reviews 2 (AMSTAR 2) [25] and consensus was reached through discussion. No reviews were excluded following quality assessment. We report overlap of eligible primary studies in a citation matrix (S2 Table) [26]. We used the Corrected Coverage Area (CCA) Index to quantify the degree of eligible primary study overlap among reviews, which was 2.6% for this overview [26]. A CCA index <5% indicates slight overlap of eligible primary studies in more than two systematic reviews.

### Data synthesis

We summarize the evidence from the included reviews in two steps (S2 Fig). First, we describe the review characteristics, authors' conclusions, and results of meta-analyses when available with reviews grouped according to the medication focus of the intervention (e.g., specific medications or general deprescribing). We also summarized results for subgroups according to participant characteristics (advanced age [≥80 years of age], dementia, frailty status, and multimorbidity), intervention type and intervention setting when a review presented direct comparisons in a meta-analysis (Step 1).

Second, we present a narrative synthesis of each outcome of eligible studies from included reviews (Step 2). Some reviews had a subset of included studies that did not meet our PICOS criteria (i.e., ineligible studies); these ineligible studies were excluded from our narrative synthesis according to outcome (S2 Fig). We categorized the results of each outcome of reviews into six mutually exclusive categories that describe the broad findings: (1) beneficial effects only, (2) both beneficial effects and no evidence of effect, (3) no evidence of effect, (4) beneficial, negative and no evidence of effect (i.e., mixed effects), (5) both negative effects or no evidence of effect and (6) negative effects only (more detail can be found in S2 Fig). Outcomes were included if statistical significance testing was performed, with beneficial effects defined as a favorable effect size compared to the control group that was statistically significant based on confidence intervals or p-values,

The data were descriptively reported in this overview; therefore, no statistical analysis was performed.

### Expert panel

We convened an expert panel of seven interprofessional members drawn from medicine, pharmacy, and nursing with specialization in geriatrics. We identified members that had clinical and/or research experience in optimizing and deprescribing medications in older adults. Panel members represent diverse practice settings (Department of Veterans Affairs, academic medical centers, ambulatory care, home visits) and specialties (oncology, internal medicine, geriatrics/palliative care). The panel advised on key steps (including the study design, interpretation and presentation of results and manuscript review). Potential conflicts of interest were assessed at the time of manuscript submission and all members of the expert panel stated they had no conflicts of interest.

## Results

### Description of review characteristics (Step 1)

A total of 5,302 articles were retrieved, resulting in 3,228 unique citations. The full text of 135 articles was screened for eligibility for inclusion (Fig 1). Reasons for exclusion at the full-text stage are detailed in S3 Table. In total, 48 reviews were included, with 17 reporting meta-analyses. Characteristics of included reviews are provided in S4 Table.

**Medication focus of deprescribing.**    Sixteen reviews focused on specific medication classes or therapeutic categories including antihyperglycemics [27, 28], anticholinergics [29, 30], antihypertensives [31], psychotropics [32–40], and proton-pump inhibitors [41], or more than one medication class [15]. Thirty-one reviews focused on general deprescribing, such as reducing PIMs or polypharmacy [13, 42–71]. One review included studies that focused on specific medication classes or therapeutic categories and general deprescribing [12].

**Settings.**    Twenty-four of the 48 reviews included studies from multiple patient settings [12, 13, 15, 27, 28, 30, 31, 33, 37, 41, 43, 48, 53, 55, 56, 59–62, 64–66, 68, 69] and 16 focused on a single setting (i.e. inpatient [44, 46, 51, 67, 71], community-dwelling/outpatient [35, 39, 42, 45, 49, 50, 57], and long-term care facilities [34, 36, 38, 70]), The remaining eight did not state the setting of interest [29, 32, 40, 47, 52, 54, 58, 63]. Three reviews focused on participants with limited life expectancy [13, 46, 65].

**Interventions.**    Most reviews included studies with various intervention types, whereas 10 reviews examined specific interventions (*e.g.*, medication withdrawal [15, 31, 32], medication review [37, 42, 57], computerized decision support [51, 61], use of specific tools [44, 66]). Overall, 14 reviews included studies with investigator-initiated medication-withdrawal interventions.

**Outcomes.**    Most (n = 27) reviews pre-specified both medication-related and *other* outcomes of interest (S4 Table) [12, 13, 29, 30, 33, 37–39, 41–46, 48, 49, 51, 52, 54–56, 61, 62, 64, 66–68]; 11 reviews pre-specified only medication-related outcomes [32, 35, 36, 40, 47, 50, 53, 58, 63, 69, 71], seven pre-specified only *other* outcomes [15, 27, 28, 31, 34, 59, 65], and three did not specify specific outcomes [57, 60, 70]. Ten reviews specified an objective of evaluating harms of deprescribing [12, 15, 27, 28, 31, 33, 45, 55, 62, 68], with six reviews defining which outcomes were considered harms [12, 31, 33, 55, 62, 68]. Nine reviews specified ADWEs, including return of original condition, as an outcome of interest [12, 31, 33, 39, 41, 45, 56, 67, 68]. Only six reviews included GRADE assessments for one or more outcomes [27, 31, 45, 48, 56, 62].

Review authors' conclusions regarding the effect of deprescribing are reported in S5 Table.

## Assessment of quality of review

Assessment with AMSTAR 2 revealed that, of the 48 included reviews, one rated 'high' [46], six rated 'low' [28, 31, 36, 40, 55, 62], and 41 rated 'critically low' on overall confidence in results. Six reviews had one critical weakness [28, 31, 36, 40, 55, 62]; 14 had two [12, 27, 29, 34, 41, 42, 50, 51, 53, 61, 63, 64, 68, 71]; and the remaining had three or more (S6 Table). Included reviews scored poorly on three proposed critical domains in particular: evaluation of search strategy (question 4), reporting of the sources of funding (question 10), and evaluation of publication bias (question 15). Two questions focused on risk of bias assessment in the reviews: 13 reviews had a weakness for risk of bias assessment, and 19 had a weakness in accounting for risk of bias in the discussion of the results.

## Summary of meta-analyses reported in included reviews (Step 1, S5 Table)

**Medication-related outcomes.**    Twelve of the 48 reviews reported meta-analyses of medication-related outcomes, and these included 33 comparisons examining discontinuation, change in medication or appropriateness [12, 32, 35, 37–39, 43, 48, 51, 53, 62, 71].

Of the 33 comparisons in the 12 reviews, 25 favored the deprescribing intervention [12, 32, 35, 37–39, 43, 48, 51, 53, 62, 71], no difference was reported for seven outcomes [12, 32, 35, 38, 53, 62], and one favored the comparison [35]. All 12 reviews reported that interventions resulted in a statistically significant improvement for at least one medication-related outcome,

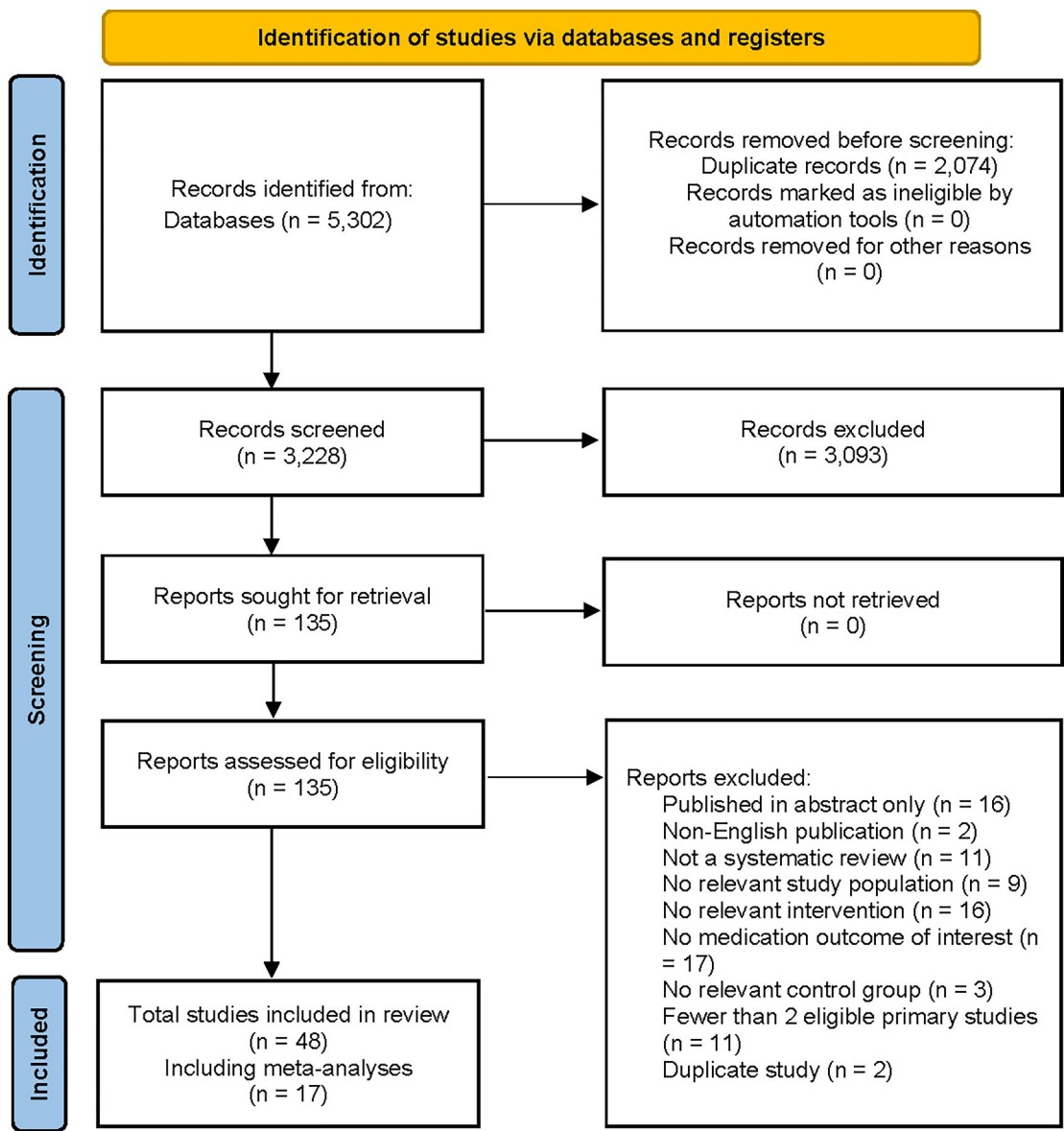

**Fig 1. PRISMA diagram summarizing number of studies identified, screened, eligible and included in final overview from literature search.**

but for some reviews reporting on multiple medication-related outcomes, no evidence of effect was found for some or results varied by intervention type [12, 32, 35, 38, 43, 53, 62]. Deprescribing interventions resulted in statistically significant reductions in number of medications [12, 53] and psychotropics [37], but not with antipsychotics [38]. Two reviews reported significant reductions for PIMs [12, 51] and two reviews with two PIM outcomes reported mixed results [53, 62]. Three reviews for benzodiazepine/hypnotic reduction reported a significant reduction [38, 39, 48], one no evidence of effect [12] and two a significant reduction for only some intervention types [32, 35]. Four reviews found a statistically significant improvement in measures of medication appropriateness [43, 53, 62, 71].

*Other* **outcomes.** Eight of the 48 reviews reported meta-analyses for *other* outcomes [12, 31, 42, 43, 45, 48, 56, 65]. For mortality, three reviews showed no evidence of effect [31, 43,

56], two reviews showed a small significant reduction [45, 65], while one review showed mixed results depending on intervention type and study design [12]. Results for the remaining outcomes that were meta-analyzed in these eight reviews included: no evidence of effect on hospitalization [31, 42, 43, 45, 48, 65], adverse drug events [43], ADWEs [12], HRQoL [31, 43], cognitive function [43], or behavioral symptoms in people with dementia [12, 43]. Mixed findings were reported for falls with three reviews reporting no evidence of effect [43, 48, 65] and one review reporting mixed results [12]. Antihypertensive medication withdrawal resulted in significantly higher blood pressure in the intervention group compared with continuation [12, 31] and higher odds of restarting antihypertensives in the intervention group [31].

**Pre-specified subgroups.** Six of the 48 reviews reported meta-analysis results for at least one of our subgroups: age [12], dementia status [12], setting [53], and intervention type (S5 Table) [12, 32, 35, 43, 45, 53]. Although other reviews were conducted in people with dementia [36], frailty [55, 64, 69], and multimorbidity [59], these are not reported in this section as there were no direct comparisons between levels within the subgroups (e.g. with and without frailty). The results according to subgroup and outcome are presented in Table 3.

*Age (1 review).* The effect of deprescribing interventions to address *polypharmacy* on mortality did not vary according to age group [12].

*Dementia (1 review).* The effect of interventions on mortality did not vary according to dementia status [12].

*Intervention type (6 reviews).* A reduction in mortality was found in randomized studies with *patient-specific* interventions but not with *educational programs* [12]. *Patient-centered* and *healthcare professional-centered* interventions were effective in reducing number of medications [53]. *Medication reviews* and *computerized decision support* improved medication appropriateness, whereas *multidisciplinary team meetings* or *staff education* were not

**Table 3. Effect of interventions according to subgroups examined in meta-analyses.**

| Subgroup | Results |
|---|---|
| Age | **Mortality**: no difference by age group [aged < 80 years (OR 0.64, CI 0.40–1.04) and those aged ≥80 (OR 0.88, CI 0.58–1.34) [12]. |
| Dementia | **Mortality**: no difference by dementia status [participants with dementia (OR 0.89, 0.63–1.27) vs. intact cognition: (OR 0.64, 0.36–1.13) [12]. |
| Intervention type | **Mortality**:<br>○ ↓ with *patient-specific* interventions (OR 0.62; 0.43, 0.88) but not with *educational programs* (OR 1.21, 0.86–1.69) [12]<br>○ ↓ with *comprehensive medication reviews* (OR 0.74, 0.58–0.95); other intervention types not analyzed [45][a]<br>○ **Number of medications:** ↓ with *patient-centered* (MD -1.01, -2.00 to -0.03) and *healthcare professional-centered* (MD -0.51, -0.80 to -0.22) interventions [53]**Medication appropriateness:** ↑ with *medication reviews* (RR 0.62, 0.41–0.93) and *computerized decision support* (RR 0.78, 0.64–0.95) but not with *multidisciplinary team meetings* (RR 0.97, 0.92–1.03) or *staff education* (RR 0.66, 0.43–1.01) [43]**Benzodiazepine use**: ↓ with some interventions (i.e. brief interventions [35], psychological interventions +/- GDR [35], cognitive behavioral therapy [32], educational programs) [32] but not others (i.e. therapeutic substitution) [32, 35].[b]**Hospitalization**: no difference with *comprehensive medication reviews* (RR 1.07, 0.92–1.26; other intervention types not analyzed [45].[a] |
| Setting | **Medication use**: ↓ in outpatient setting (MD -0.80, -1.40 to -0.21) but not in hospital setting (MD 0.50, -1.36 to 0.37) [53]. |

ADRs, adverse drug reactions; GDR, gradual dose reduction; HRQoL, health-related quality of life;

MAI = medication appropriateness.

[a] Comparison was not made with educational interventions because of heterogeneity, but no evidence of effect with education interventions on mortality or hospitalization.

[b] Refer to S5 Table for effect sizes.

associated with significant differences in this outcome [43]. Only some interventions to reduce benzodiazepine use were effective [32, 35]. *Comprehensive medication reviews* significantly reduced mortality but had no evidence of effect on hospitalizations, but comparison was not made with other interventions due to heterogeneity (educational interventions) or studies did not report on these outcomes (computerized decision support) [45].

*Setting (1 review)*. Interventions resulted in lower medication use in the outpatient setting but not in the hospital setting [53].

## Narrative synthesis by outcomes at the systematic review level (Step 2, Table 4)

Table 4 summarizes evidence from 43 reviews, by outcomes, based on six mutually exclusive categories of intervention effects. Five reviews did not contribute data to this table because

**Table 4. Summary of deprescribing interventions for outcome themes according to medication focus of the review (Step 2)[a].**

| Evidence suggested: | A Beneficial effect only | B Beneficial and no effect | C No effect | D Beneficial, no effect, and negative effect | E No effect and negative effect | F Negative effect only |
|---|---|---|---|---|---|---|
| **MEDICATION-RELATED OUTCOMES** | | | | | | |
| Medication reduction | 13,*27,33,34,38,41*,42,49,52,55,68 | *29,32,36*,44–48,50,51,54, 56–61,63,64, 67,69,70 | | 66 | | |
| Medication appropriateness | *29*,42,48–50, 56–58,64, 67,70 | *30*,47,66 | | | | |
| *OTHER* OUTCOMES | | | | | | |
| Surrogate biomarkers | | | *27,28*,49 | | 68 | *15* |
| Mortality | | 13,68 | *27,28,37,38*,42,44,45,49,51,52,55,60,61,64,67,70,71 | | | |
| HRQoL | | 13,*33*,43,62,67 | *31,34,37*,42,45,47,55,60 | 48,56 | 50,57 | |
| Patient perception-treatment burden | 57 | | 55 | | | |
| Cognition | 55 | *34* | 13,*29,33,38*,56,70 | | | |
| Falls | 51 | 13,45,57,66, 67,70 | *34,37*,44,46–48,54,55 | | | |
| Hospitalizations | 47 | 56,62,66,71 | *34,38*,42,44,45,48,51,52,55,57,61,64,65,67,70 | | | |
| Costs | | | | | | |
| Adverse events[b] | *29,31,37* 55, 58 | 47,57,62,67 | *32,38* 42,43,46,50,56,68,70 | | | *33* 68,71 |
| Other PROMs[c] | 44,47,55,66,70 | | *27,28,34,37,38* 13,47,56,67,70 | *33,37* | *34* | *34,38* |

HRQoL, health-related quality of life; PROMs, patient-reported outcome measures.

[a]Bolded citations distinguish reviews that focused on specific medication classes or therapeutic categories from those that focused on general deprescribing. Only results for outcomes of eligible primary studies of included reviews with statistical significance testing reported were included.

[b]Adverse events include adverse drug events/ADRs and ADWEs. Beneficial effects refer to reduced adverse events, while negative effects refer to increased adverse events. Studies with ADWEs: **Col C** [43]; **Col F** [33, 68]

[c]**Col A**: pain [47, 70], Activities of Daily Living [44], functional status [55], neuropsychiatric symptom score [66]. **Col C**: Activities of Daily Living [34], mobility [47, 70], hypoglycemia-but was not classified as an adverse event so reported here [27, 28], mobility [47, 70], confusion [47, 70], behavior/agitation [34, 38, 47, 70], urinary incontinence [47], depressive symptoms [37, 47, 70], function [13, 37, 47, 56, 67], sleep [13, 34], performance status, bowel, and symptom status [13]. **Col D** sleep quality [33], behavioral and psychological symptoms of dementia [37]; **Col E**: depression [34]; **Col F**: depression [38], apathy and psychiatric symptoms [34].

only meta-analysis results were reported for included outcomes [12, 35, 39, 53] or the effect of interventions on outcomes was not clear from the review synthesis [40]. Bolded citations distinguish reviews that focused on specific medication classes or therapeutic categories from those that focused on general deprescribing. The most common outcomes reported were medication reduction (n = 34) [13, 27, 29, 32–34, 36, 38, 41, 42, 44–52, 54–61, 63, 64, 66–70], adverse events (n = 21: adverse drug reactions (ADRs), n = 18, [29, 31, 32, 37, 38, 42, 46, 47, 50, 55–58, 62, 67, 68, 70, 71]; ADWE, n = 3, [33, 43, 68]), hospitalizations (n = 20) [34, 38, 42, 44, 45, 47, 48, 51, 52, 55–57, 61, 62, 64–67, 70, 71], mortality (n = 19) [13, 27, 28, 37, 38, 42, 44, 45, 49, 51, 52, 55, 60, 61, 64, 67, 68, 70, 71], HRQoL (n = 17) [13, 31, 33, 34, 37, 42, 43, 45, 47, 48, 50, 55–57, 60, 62, 67], falls (n = 15) [13, 34, 37, 44–48, 51, 54, 55, 57, 66, 67, 70], medication appropriateness (n = 14) [29, 30, 42, 47–50, 56–58, 64, 66, 67, 70], and cognition (n = 8) [13, 29, 33, 34, 38, 55, 56, 70]. No reviews contributed information for healthcare costs.

**Medication-related outcomes.** *Medication reduction (34 reviews).* Most reviews (33 of 34) reported a reduction in the number of medications, target medication or PIMs: 11 reported beneficial effects only (Col A, Table 3) and 22 reported both beneficial effects and no evidence of effect (Col B). One (of 34), reported mixed effects (Col D).

*Medication appropriateness (14 reviews).* All reviews (14 of 14) reported some beneficial effects; 11 reported beneficial effects only (Col A) and three reported both beneficial effects and no evidence of effect (Col B).

*Specific medication class or therapeutic category (9 reviews; 10 medication-related outcomes).* Of the 9 reviews that focused on a single medication/class target (Table 3, bolded and italicized references), 10 medication-related outcomes were reported (six with beneficial effects only [Col A] and four with both beneficial effects and no evidence of effect [Col B]). Five of these nine reviews focused on psychotropics (S7 Table); three reported beneficial effects only (Col A) and two reported both beneficial effects and no evidence of an effect (Col B).

*General deprescribing (26 reviews, 38 medication-related outcomes).* Of the 26 reviews that included studies of general deprescribing, 38 medication-related outcomes were reported: 16 with beneficial effects (Col A), 21 with both beneficial effects and no evidence of effect (Col B), and one with mixed effects (Col D).

**Adverse events and *other* outcomes.** *Adverse events (20 reviews, 21 outcomes [18 ADRs and 3 ADWEs]).* 9 of 18 reviews reported some reduction in ADRs (beneficial effects); 5 reported reductions only (Col A) and four reported both reductions and no evidence of effect (Col B). Eight of 18 reviews reported no evidence of an effect on ADRs (Col C), and one review reported an increase in ADRs (negative effect only, Col F). Three reviews reported on ADWEs: one review reported no evidence of an effect (Col C) and two studies reporting an increase in ADWEs (negative effect only, Col F).

*Hospitalizations (20 reviews).* The results for the 20 reviews include: one review reported a beneficial effect (Col A), four reviews reported both beneficial effects and no evidence of effect (Col B) and 15 reported no evidence of effect only (Col C).

*Mortality (19 reviews).* The results for the 19 reviews include: two reviews reported both beneficial effects and no evidence of effect (Col B), and 17 reported no evidence of effect only (Col C).

*HRQoL (17 reviews).* The results for the 17 reviews include: five reviews reported both beneficial effects and no evidence of effect (Col B), eight reported no evidence of effect only (Col C), two reported mixed effects (Col D), and two reported no effect and a negative effect (Col E).

*Falls (15 reviews).* The results for the 15 reviews include: one review reported a reduction in falls (Col A), six reviews reported both beneficial effects and no evidence of effect (Col B), and 8 reported no evidence of effect only (Col C).

Other negative effects (Cols E and F) were found for surrogate biomarkers (n = 2) and other patient-reported outcome measures (n = 3).

## Discussion

Deprescribing medications is a priority for older patients to address polypharmacy and enhance quality of prescribing [72, 73]. This is the first overview of systematic reviews examining deprescribing interventions in older adults with the goal of synthesizing the breadth of research available, elucidating key findings, and identifying gaps and future research priorities. We summarized results from meta-analyses from included reviews and performed a narrative synthesis according to outcome. In our narrative synthesis, we found that deprescribing interventions generally reduced medication use, but results for *other* outcomes were either mixed or there was no evidence of an effect. While adverse events were reported in nearly half (42%) of reviews, only three reviews reported that the intervention increased adverse events, two of which were ADWEs. Few reviews reported meta-analyses for *other* outcomes and those that did reported either mixed effects (*e.g.*, mortality, falls) or no evidence of an effect (*e.g.*, hospital admissions, adverse events, HRQoL, cognitive function). Few reviews compared whether the outcomes varied according to patient characteristics such as extremes of older age, dementia, frailty or multimorbidity. Although reviews did report on various costs, no reviews reported costs for hospitalization or emergency department visits.

Our findings align with prior research [12] and widespread consensus [74] that deprescribing interventions in older adults are likely to be successful in medication reduction and do not appear to increase adverse outcomes, though their effect on adverse outcomes is based on lack of evidence and our overview has found that ADWEs are not frequently reported.

This overview of reviews highlights several evidence gaps that should be addressed to advance the uptake of deprescribing in clinical practice. First, we found that the majority of deprescribing reviews did not specifically aim to examine harms, such as ADWEs, including the return of the original symptom/condition. Defining potential harms of deprescribing will be important to facilitate increased uptake of deprescribing by patients and providers [75, 76]. Healthcare professionals have identified safety concerns as a barrier for deprescribing [77, 78]. Future studies should examine harms as a primary outcome and clearly specify which outcomes are considered harms of deprescribing [79]. Return of the symptoms/condition following deprescribing varies according to medication class and underlying condition being treated, ranging from a recurrence of gastrointestinal reflux symptoms to a recurrence of more serious conditions such as heart failure or severe depression [80]. Researchers should select ADWEs as outcomes based on what would be expected for the medication target (*e.g.*, benzodiazepines, antihypertensives) and use standard methods for the detection of and causality assessment [81].

Second, few reviews directly compared outcomes according to intervention type. Limited evidence suggests that some interventions (*e.g.*, patient-specific, patient-centered, medication review, computerized clinical support) were more effective than others, however, interpretation of these comparisons was difficult due to a lack of a standard taxonomy of deprescribing interventions. Reviews provided minimal description of the original interventions, which may have been a limitation of poor reporting of interventions in the original studies.

Third, although interventions led to medication reduction, we identified a dearth of research on *other* downstream outcomes. We were able to extract data from 31–42% of reviews for mortality, HRQoL, falls and adverse events, however only 17% of reviews contributed information on cognition. Where reviews did report on these outcomes, few found evidence of benefit. The fact that many reviews found no evidence of effect for these outcomes is not

surprising given the methodological challenges faced in deprescribing-intervention trials. These *other* outcomes are dependent on the success of the interventions on medication discontinuation, but the effect sizes of deprescribing interventions on medication discontinuation are often small. Furthermore, studies are under-powered and have insufficient follow-up periods to find differences in these *other* outcomes, which are often included as secondary outcomes. Additionally, it has been identified that many deprescribing reviews include studies where deprescribing is not the only focus of the intervention, often including the aim to prescribe appropriate medications or reduce initiation of inappropriate medications. This may dilute impact on *other* outcomes [82]. Deprescribing research would benefit from the selection and reporting of consistent outcomes and the development of a core outcome set specific to deprescribing [83]. It is important to measure patient-reported outcomes, including the satisfaction with and values around medication discontinuation and how it impacts patient experience of care. Although robust information about whether interventions reduce healthcare costs was lacking in our synthesis, one included review examined the economic impact of deprescribing and concluded that the evidence is limited to determine whether the benefits of deprescribing outweigh implementation costs [59].

Finally, there was limited evidence focused on deprescribing in vulnerable subgroups, including patients with dementia, frailty and multimorbidity. Interestingly, two rigorous deprescribing trials were recently published and focused on patients with dementia [84] or multimorbidity [80], suggesting that researchers are starting to address this gap in these vulnerable patients. Additional research is needed for these subgroups as the effect of interventions might differ according to patient characteristics. Frail older adults with diabetes and hypertension are more likely to have adverse effects from treatment [85–87], while those with dementia might be more sensitive to psychotropic medications [88]. Thus, interventions could have a greater impact in these groups to reduce adverse drug events. Given that the main barriers to deprescribing in practice are time and resources, healthcare decision makers and clinicians are looking to identify individuals who may benefit most from deprescribing efforts so that resources can be effectively allocated.

## Strengths and limitations

We conducted a comprehensive overview of systematic reviews using established methods [16]. Of the 48 included reviews, 40 (83%) were published since 2016. This overview of reviews fills an important gap by summarizing the proliferation of evidence from recent deprescribing reviews. Although a core outcome set was not available for deprescribing, we adapted a core outcome set for polypharmacy research to conceptualize the results for this overview [23].

Despite these strengths, we acknowledge several limitations of this overview. Given the nature of an overview, we did not capture large recently published trials that have focused on deprescribing interventions [80, 84, 89]. Two of three of these trials examined the effectiveness on downstream outcomes—even as primary outcomes [80, 89]. Only one of these trials reported a reduction in medications, but no difference in adverse drug events between groups [89]. We included for synthesis only English-language publications due to lack of proficiency with other languages–although we note that only one review was excluded for this reason [90]. We aimed to explore the effect of deprescribing in older adults and there was some variation on how this was reported so we used an age cut off point of 60 years to ensure we included as many reviews of older adults as possible. Further, as an overview of reviews, consistent with prior established methods [16], we did not search for, extract directly from, or assess the quality of the original primary studies. Most reviews were of low or critically low quality, however, a 'floor effect' of AMSTAR 2 has been acknowledged [91, 92]. Our information was dependent

on reporting by authors of the reviews, which varied in quality and style. Although we assessed risk of bias as part of the AMSTAR 2 tool for each review, we did not extract information on individual included studies, though this has been recommended in a recent reporting guideline published after we completed our overview [93]. We summarized GRADE assessments for certainty of evidence for outcomes presented in reviews, however, few reviews included this information, and it would not have been appropriate for us to conduct our own GRADE assessments of primary studies in reviews. Finally, as an overview of reviews, we were only able to extract outcomes reported in included reviews regardless of whether they reported both the impact on medication-related outcomes and other downstream important clinical outcomes. We were unable to estimate the association between success on medication-related outcomes and its impact on *other* outcomes.

## Implications for policy and research

We have identified five gaps in the current evidence that are priorities for future research: 1) define and examine potential harms of deprescribing; 2) develop a taxonomy of intervention types; 3) develop and use a standardized set of outcome measures, which include patient-reported measures, clinical, health utilization and cost outcomes and report on both the clinical and statistical significance for effects; 4) conduct intervention studies of sufficient sample size and duration to be able to capture such outcomes to inform practice and policy; and 5) examine the effect of deprescribing on specific vulnerable subgroups, such as people at the extremes of age or those with frailty, dementia, or multimorbidity. Leveraging the multidisciplinary groups within nascent deprescribing networks around the world (i.e., Canada, US, Europe, UK and Australia) [94], will be crucial in catalyzing the development of such research, addressing these research gaps, and translating findings into clinical practice.

## Conclusions

In summary, interventions with a deprescribing focus generally resulted in medication reduction. Information about *other* outcomes was not routinely examined, and where included, studies were likely underpowered. Although the reviews were mostly of low quality, the evidence suggests that deprescribing was likely not associated with increased adverse events, and had little evidence of effect on mortality, HRQoL and health-care use. Few reviews examined effects on vulnerable subgroups. Nonetheless, the evidence is clear that polypharmacy and use of high-risk medications can result in patient harm. Clinicians should continue to look for opportunities to deprescribe inappropriate medications and practice shared decision-making, keeping in mind patient-specific goals in deprescribing medications. Even modest reductions in medicines can be beneficial to individual patients and will have wider impacts at a population level in terms of overall harm reduction and costs of care delivery.

## Supporting information

**S1 Checklist. PRISMA 2020 checklist.**
(DOCX)

**S1 Appendix. Search strategy and databases.**
(DOCX)

**S1 Fig. Conceptual model.**
(PDF)

**S2 Fig. Example of data abstraction for narrative synthesis of eligible studies of systematic reviews for two outcomes (Step 2, Table 4).**
(DOCX)

**S1 Table. Amendments to protocol.**
(DOCX)

**S2 Table. Citation matrix for eligible primary studies within included systematic reviews.**
(XLSX)

**S3 Table. List of articles excluded at full-text, with reasons.**
(DOCX)

**S4 Table. Study characteristics of included systematic reviews.**
(DOCX)

**S5 Table. Summary of authors' conclusions from systematic reviews and results of meta-analyses (Step 1).**
(DOCX)

**S6 Table. AMSTAR 2 quality assessment of included systematic reviews.**
(DOCX)

**S7 Table. Summary of the effect of deprescribing interventions on outcome themes, grouped by specific medication classes.**
(DOCX)

## Acknowledgments

We would like to acknowledge the expert panel members as authors that provided input during the design phase, interpretation of results and reviewed an early and final draft. Affiliations are at the time the work was conducted. The expert panel includes: Kenneth Boockvar, MD, James J. Peters Veterans Affairs Medical Center, Bronx, NY; Holly Holmes, MD, MS, University of Texas MD Anderson Cancer Center, Houston, TX; Jean Kutner, MD, MSPH, University of Colorado School of Medicine, Denver, CO; Sunny Linnebur, PharmD, CGP, University of Colorado School of Pharmacy, Denver, CO; Zachary Marcum, PharmD, PhD, University of Washington School of Pharmacy, Seattle, WA; Elizabeth Phelan, MD, MS, University of Washington School of Medicine, Seattle, WA; Marianne Shaughnessy, PhD, CRNP, US Department of Veterans Affairs, DC; Sarah Szanton, Ph.D. Johns Hopkins University School of Nursing. The panel members reported no conflicts of interest. We thank Naomi Schwartz and Yuhan Huang, PhD candidate for assisting with screening of titles and abstracts and data extraction. We would also like to thank Diana Louden, MS for developing and conducting the search strategy.

## Author Contributions

**Conceptualization:** Beth Devine, Shelly L. Gray.

**Data curation:** Julia Fox, Shelly L. Gray.

**Funding acquisition:** Beth Devine, Shelly L. Gray.

**Methodology:** Adam Todd, Emily Reeve, Susan M. Smith, Stephen Hughes, Andrew Husband, Beth Devine, Shelly L. Gray.

**Project administration:** Shelly L. Gray.

**Resources:** Emily Reeve, Susan M. Smith, Jeffrey R. Harris, Shelly L. Gray.

**Supervision:** Shelly L. Gray.

**Validation:** Shiyun Chua, Zizi Elsisi, Stephen Hughes, Aili Langford, Niamh Merriman.

**Visualization:** Shiyun Chua, Adam Todd, Emily Reeve, Susan M. Smith, Beth Devine.

**Writing – original draft:** Shiyun Chua, Adam Todd, Shelly L. Gray.

**Writing – review & editing:** Shiyun Chua, Adam Todd, Emily Reeve, Susan M. Smith, Julia Fox, Zizi Elsisi, Stephen Hughes, Andrew Husband, Aili Langford, Niamh Merriman, Jeffrey R. Harris, Beth Devine, Shelly L. Gray.

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
