## [Decision Letter · Decision Letter 0]

30 Jan 2024

PONE-D-23-42998Effectiveness of Deprescribing Interventions in Older Adults: An Overview of Systematic ReviewsPLOS ONE

Dear Dr. Gray,

Thank you for submitting your manuscript to PLOS ONE. After careful consideration, we feel that it has merit but does not fully meet PLOS ONE’s publication criteria as it currently stands. Therefore, we invite you to submit a revised version of the manuscript that addresses the points raised during the review process.

**Dear Respectable Authors**Please respond to all comments in a clear and point-by-point manner, taking into account the opinions of the respected judges.

Cheers

We look forward to receiving your revised manuscript.

Kind regards,

Morteza Arab-Zozani, Ph. D.

Academic Editor

PLOS ONE

Reviewers' comments:

Reviewer's Responses to Questions

**Comments to the Author**

1. Is the manuscript technically sound, and do the data support the conclusions?

Reviewer #1: No

Reviewer #2: Yes

Reviewer #3: Partly

2. Has the statistical analysis been performed appropriately and rigorously? 

Reviewer #1: No

Reviewer #2: N/A

Reviewer #3: N/A

3. Have the authors made all data underlying the findings in their manuscript fully available?

Reviewer #1: Yes

Reviewer #2: Yes

Reviewer #3: Yes

4. Is the manuscript presented in an intelligible fashion and written in standard English?

Reviewer #1: Yes

Reviewer #2: Yes

Reviewer #3: Yes

5. Review Comments to the Author

Reviewer #1: Thank you very much for the opportunity to review the interesting manuscript. The topic, an overview of systematic reviews on effects of deprescribing interventions in older adults, is interesting. Please see my comments below.

Major revision

This is an important topic, but the focus is very broad and the manuscript did not sufficiently

take into account the complexly of the field. A large number of different intervention approaches have been evaluated for deprescribing medications in older adults, but the paper does not distinguish between the different types of intervention. Although it is mentioned that “we synthesized results according to the focus of the intervention (i.e., deprescribing specific medication targets, or general deprescribing)” (line 111), this was done as a subgroup analysis and did not really address different types of interventions. The focus on outcomes irrespective of the intervention approach seems too simple to draw clinically relevant conclusions. The methodological quality of the reviews and the certainty of evidence were also not taken into account in the analysis and the interpretation of the results. All except one of the included reviews had an AMSTAR 2 rating of low or critically low. Methodological guidance on conducting systematic reviews of complex interventions was also not incorporated in this review (e.g. Guise JM et al. 2017. AHRQ series on complex intervention systematic reviews-paper 1: an introduction to a series of articles that provide guidance and tools for reviews of complex interventions. J Clin Epidemiol 2017;90:6-10. doi: 10.1016/j.jclinepi.2017.06.011). Therefore, the main conclusion “Deprescribing interventions likely resulted in medication reduction” seems not appropriate.

Detailed comments

Abstract

Please specify the objective.

I recommend avoiding the term “effectiveness”, since the manuscript do not clearly distinguish between studies evaluating efficacy and effectiveness. The term “effects” seems to fit better.

Methods - “We included randomized or non-randomized controlled designs.” Please revise the sentence, since reviews and not primary studies were included.

Introduction

Although some approaches for deprescribing interventions in older adults are mentioned, detailed information are lacking. Different intervention approaches use different mechanisms to change prescribing practice. A subgroups analysis based on intervention type is mentioned in the introduction, but not presented in detail in the manuscript. No rational for this unspecific review question is presented.

Methods

The methods seem appropriate with exception of the analysis and synthesis (see my general comment above). However, such an intervention-centred approach seems crucial for such an overview of complex interventions.

Line 157: “(…) their GRADE [24] assessment of study quality (…)”: GRADE us used to assess the quality or certainty of the evidence on a specific outcome rather than the study quality. Please revise.

Data synthesis

Please describe in detail how “beneficial effects” were defined, e.g. based on effect estimates or confidence intervals (CI)? Was the precision of the results considered and the quality/certainty of evidence for the respective outcome? If quality/certainty of evidence was considered, how about reviews that did not use the GRADE approach?

Please describe in more detail who was part of the expert panel (including information of methods to handle conflicts of interests) and how the expert panel contributed to the review? Was a consensus process used or similar approaches?

Results

As mentioned above, the summary of the results is too superficial and lacks details about the intervention approaches, intervention details (e.g. components) and detailed information about the study results, certainty of evidence and the methodological quality of the reviews.

The information on the different intervention types (line 363 and following) does not report sufficient details about the intervention approaches.

Reviewer #2: anuscript ID: PONE-D-23-42998

Effectiveness of Deprescribing Interventions in Older Adults: An Overview of Systematic Reviews

The authors have synthesized evidence from reviews to provide an overarching overview of effectiveness of Deprescribing interventions in older adults. This is a valuable piece of work and appreciate the authors’ effort. Some of my comments are as follows:

1. Introduction – L83: The authors state that earlier work focused on reducing polypharmacy by addressing the appropriateness of medications and discontinuing those where harms outweigh benefits. As a reader, I would want to look into some of these studies. The authors can highlight these studies providing reference(s).

2. Method – Search strategy - L120: The term information specialist might not be a common word in some parts of the world. It might be worth defining it or providing alternative word in a bracket such as ‘librarian’ for wider audience.

3. Method – Table 1 – Population: I appreciate that the authors have considered studies conducted in long-term care even if age was not provided. My comment is related to the definition of population. The authors have considered aged 60 years and above as older adults. However, the authors have used search term ‘Aged’ and ‘older adults’ in their search strategy, which are considered as electronic databases as person 65 years and above. Often studies on older adults have considered the later definition. So, it would be great if the authors could clarify this.

4. Method - Expert panel: The authors have considered an interprofessional expert panel for providing input in this review. I understand that they had a broad range of clinical and perspective related to the care of older adults. But for the methodological rigor the reader would benefit from knowing about the specific criteria for their inclusion.

5. Result – Step 2 (page 12, L276): The author state there were 19 studies reporting on mortality but have cited only 18 studies.

6. Discussion – please consider the grammar – L384: “…and those that did found…”

7. Discussion: The authors have raised a really good point that majority of reviews did not specifically aim to examine harms. While this is true, studies reporting on the causality of harm due to deprescribing is also limited. It is also true that unless evidence of harm associated with deprescribing is well established, deprescribing option should be open to patients. This is not a comment but a view.

Reviewer #3: Thank you for asking me to review this manuscript. This topic is important and timely. There are some comments below that I think should be seriously considered to ensure this overview meets current standards and provides more information to the reader.

Comments:

Major:

a. Step 1: summarizing conclusions of reviews and meta-analyses: Although the authors have charted the review authors’ conclusions (in a supplement table), they do not perform any synthesis on the conclusions or even describe them at any length. I would suggest not considering these review conclusions as part of the overview’s synthesis and just refer to this information as charted in the appendix.

b. A key step in an overview (as per Cochrane ch 5 guidance) is to collect, analyze and present the risk of bias of the included studies (usually as assessed by the review authors), and to integrate this into the overview’s synthesis. If ROB assessments are not complete or are inadequate, some overview authors will re-assess the studies themselves. The new reporting guidance for overviews (Gates et al. PRIOR) asks that if authors do not re-asses any of the primary studies (when identifying flaws) that they provide justification for this.

c. The synthesis for steps 1 and 2 currently focuses on a count of reviews that had particular conclusions about the interventions, of which very few assessed the certainty of evidence (eg GRADE) to help the authors describes to what degree factors such as directness, ROB, imprecision may have impacted the conclusions. Without the ability to rely on and add the review authors’ certainty assessments, the overview authors should (if not willing to perform GRADE) add some information about these factors and consider whether any of these factors changes the conclusions. For example, 2 reviews of several large well-done RCTs that found no difference for an outcome may have had higher certainty evidence than 7 reviews with few small non-randomized studies that found beneficial effects. For both synthesis steps, adding information into the main manuscript (e.g. text for meta-analysis findings and in Table 3 for each category of magnitude for each outcome) about the number of studies, total sample size, and the study ROB (e.g. % studies with high ROB) for each review outcome would add some additional insight to the findings across the categories of effect. The reviews used for these conclusions could be cited rather than the focus of the synthesis. Further, separating the effects (and data on other factors eg sample size) by general deprescribing versus specific targets (and mentioning the targets) would be informative. Table 3 could add a lot more information to be very useful for readers. An overview would ideally provide a GRADE certainty assessment about each outcome but at least providing some additional data beyond just the number of reviews to help draw conclusions would be very useful.

d. Because there was very little overlap across the reviews, it appears suitable to consider combining the results from reviews with and without meta-analysis. Results from meta-analyses can easily be categorized into the 6 categories of effect and without this added information the synthesis in step 2 appears incomplete. Step 2 could therefore combine findings across all reviews, while possibly removing reviews if their studies are in complete overlap with others reviews (especially the ones that did meta-analysis). Table 3 could be split into 2 tables if necessary; Table 2 data could likely be integrated into Table 1 easily to reduce the number of tables.

e. It appears that the categories of effect size currently focus on statistical significance and this should be clearly stated or changed to “clinically important” etc) with use of some threshold of effect. Possibly for those effects that are of “no difference” the authors could highlight the ones where the point estimate appears large (eg relative risk of 2+) but just did not reach significance? There is also the possibility to conclude “beneficial effect” when the results are statistically but not apparently of clinical importance (e.g. a precise estimate RR 1.09) but if the authors choose not to do this they should at least mention this possibility and that focus on significance can be misleading.

Minor comments:

• Abstract: We included data in reviews from randomized or non-randomized controlled designs.- original sentence made it sound as though these studies were being included vs only review of the studies

• Methods: data extraction. The screening process appeared to require 2 people to agree to progress a citation to full text, which may have led to missed studies. If there is no consensus at this stage it would better to include to full text any citation either of 2 reviewers thought should be looked at closer. This is the main reason we use 2 reviewers at this stage. Suggest to add this as a potential limitation ot the review.

• p. 7 GRADE does not assess study quality but rather the certainty of the evidence across studies (for an outcome)

• p.8 description of the categories of effect, I think (2) should be “both beneficial effects and no evidence of effect”

• p.17 the use of review authors names and underlining contrasts with the general format of the rest of the manuscript -suggest to make this consistent

6. PLOS authors have the option to publish the peer review history of their article (what does this mean?). If published, this will include your full peer review and any attached files.

Reviewer #1: No

Reviewer #2: **Yes: **Shakti Shrestha

Reviewer #3: No

---

## [Author Response · Author response to Decision Letter 0]

2 Apr 2024

Thank you for this opportunity to respond to the reviewers’ comments. We appreciate the thorough review provided. We believe that revising the manuscript based on this input has increased the quality of the manuscript. As requested, the reviewers’ are able to view the changes in the tracked change version.

Reviewer #1: 

1a. This is an important topic, but the focus is very broad and the manuscript did not sufficiently take into account the complexly of the field. A large number of different intervention approaches have been evaluated for deprescribing medications in older adults, but the paper does not distinguish between the different types of intervention. Although it is mentioned that “we synthesized results according to the focus of the intervention (i.e., deprescribing specific medication targets, or general deprescribing)” (line 111), this was done as a subgroup analysis and did not really address different types of interventions. The focus on outcomes irrespective of the intervention approach seems too simple to draw clinically relevant conclusions.

Response: The reviewer raises many excellent points. Some of these issues are difficult to address given the nature of an overview of systematic reviews. The objective of our work was to synthesize review results by outcome of interest and medication focus of intervention (i.e., specific medication classes/therapeutic categories or general describing). We chose the overview design to summarize and, where possible, synthesize the breadth of research available in the deprescribing field, elucidate key findings, and identify gaps. We sought to answer the question, when deprescribing occurs, how does this impact outcomes? We did not aim to establish the effectiveness of deprescribing interventions according to intervention type. 

We agree that summarizing outcomes according to intervention is important and believe that a detailed analysis of outcomes by intervention type would be best conducted at the level of a systematic review when authors have primary study intervention data available to them. We did report on intervention type when direct comparisons were made among interventions within a meta-analysis. To clarify our objective, we have made the following revision: 

• In abstract: “The objective of this overview of systematic reviews was to summarize the review evidence for deprescribing interventions in older adults”. 

• Line 105-111: “Our objective was to summarize the review evidence for deprescribing interventions in older adults. We sought to synthesize review data according to the medication focus of the intervention (i.e., specific medication class or therapeutic category versus general deprescribing) by outcomes of interest. In addition, we summarized review findings by pre-determined subgroups (patient characteristics, intervention type and setting) when direct comparisons were available within reviews.

• Table 3: We streamlined Table 3 to align with the clarified objective and only included reviews that made direct comparisons for the subgroups (page 13).

1.b The methodological quality of the reviews and the certainty of evidence were also not taken into account in the analysis and the interpretation of the results. All except one of the included reviews had an AMSTAR 2 rating of low or critically low. Methodological guidance on conducting systematic reviews of complex interventions was also not incorporated in this review (e.g. Guise JM et al. 2017. AHRQ series on complex intervention systematic reviews-paper 1: an introduction to a series of articles that provide guidance and tools for reviews of complex interventions. J Clin Epidemiol 2017;90:6-10. doi: 10.1016/j.jclinepi.2017.06.011). Therefore, the main conclusion “Deprescribing interventions likely resulted in medication reduction” seems not appropriate.

Response: We agree with the reviewer that the overall quality of included reviews, based on applying the AMSTAR-2 criteria was mainly low to very low. This is presented in the results section (lines 236-244) and S6 Table. We appreciate the suggestion to apply the methods guidance promulgated by AHRQ in their complex intervention series. This was, indeed, important work. After reading their work, we understand it refers to systematic reviews rather than overviews of systematic reviews, so we believe that our use of AMSTAR-2 is more appropriate. We suggest that our cautious framing of results reflects the overall low quality of included reviews. 

Detailed comments

2. Abstract

Please specify the objective.

I recommend avoiding the term “effectiveness”, since the manuscript do not clearly distinguish between studies evaluating efficacy and effectiveness. The term “effects” seems to fit better.

Response: Thank you for this suggestion. We have revised this in the title and throughout the manuscript as suggested.

3. Methods - “We included randomized or non-randomized controlled designs.” Please revise the sentence, since reviews and not primary studies were included.

Response: We have revised this as suggested. 

4. Introduction

Although some approaches for deprescribing interventions in older adults are mentioned, detailed information are lacking. Different intervention approaches use different mechanisms to change prescribing practice. A subgroups analysis based on intervention type is mentioned in the introduction, but not presented in detail in the manuscript. No rational for this unspecific review question is presented.

Response: We agree with the reviewer’s comment above about the numerous approaches for deprescribing medications. As noted in response 1, we only summarized outcomes according to interventions when direct comparisons were available within the reviews. We revised our aim to clarify this distinction. Only six of the forty-eight reviews compared one or more interventions.

5. Methods

The methods seem appropriate with exception of the analysis and synthesis (see my general comment above). However, such an intervention-centred approach seems crucial for such an overview of complex interventions.

Response: Please see response #1 and #4 for our explanation on the focus of our review and how we have clarified our aim.

6. Line 157: “(…) their GRADE [24] assessment of study quality (…)”: GRADE us used to assess the quality or certainty of the evidence on a specific outcome rather than the study quality. Please revise.

Response: Thank you for pointing this out. We have made the revision. “….assessment of quality of evidence across studies for the outcome” (Line 155)

7. Data synthesis

Please describe in detail how “beneficial effects” were defined, e.g. based on effect estimates or confidence intervals (CI)? Was the precision of the results considered and the quality/certainty of evidence for the respective outcome? If quality/certainty of evidence was considered, how about reviews that did not use the GRADE approach?

Response: Thank you for your suggestion on the need for further clarification. The effects were based on effect size, p-values and confidence intervals. The AMSTAR-2 (domain 9) has a specific question about the risk of bias of for SRs, and if a review is considered to be “weak” in this domain, the overall AMSTAR rating would automatically be low based on this criterion alone. 13 reviews were judged to have a weakness in this domain and 4 had a partial weakness. We include GRADE when available from the review (please see S4 Table, outcomes column). Only 6 reviews included GRADE for one or more of their outcomes. Given this, we did not take into account the quality of evidence for the outcome when determining the benefit. Our focus was to identify gaps and policy implications. We have added the following sentence to the results to report the number of reviews that applied GRADE (line 228):

 “Only 6 reviews included GRADE assessments for one or more outcomes [27,31,45,48,56,62]”

8. Please describe in more detail who was part of the expert panel (including information of methods to handle conflicts of interests) and how the expert panel contributed to the review? Was a consensus process used or similar approaches?

Response: The expert panel functioned as an advisory panel only. All decisions were made by the study authors. The expert panel members were asked to provide a declaration of interest and this resulted in no conflicts of interest being declared. Consensus processes were not required for the expert panel.

Please see the revised section below (lines 189-196)

“We convened an expert panel of seven interprofessional members drawn from medicine, pharmacy and nursing with specialization in geriatrics. We identified members that had clinical and/or research experience in optimizing and deprescribing medications in older adults. Panel members represent diverse practice settings (Department of Veterans Affairs, academic medical centers, ambulatory care, home visits) and specialties (oncology, internal medicine, geriatrics/palliative care). The panel advised on key steps (including the study design, interpretation and presentation of results and manuscript review). Potential conflicts of interest were assessed at the time of manuscript submission and all members of the expert panel stated they had no conflicts of interest.

9. Results

As mentioned above, the summary of the results is too superficial and lacks details about the intervention approaches, intervention details (e.g. components) and detailed information about the study results, certainty of evidence and the methodological quality of the reviews.

The information on the different intervention types (line 363 and following) does not report sufficient details about the intervention approaches.

Response: The objectives of this overview of reviews were to identify gaps and implications for policy and research. We agree that grouping/ differentiating outcomes according to intervention and summarizing the findings is very important. However, a detailed analysis of outcomes by intervention type would be best conducted at the level of a systematic review when authors have available primary study intervention data. The benefit of categorizing primary studies according to intervention components from the information available in the reviews is less clear. Please refer to Responses 1 and 4 for a more detailed explanation of our approach.

Reviewer #2: Manuscript ID: PONE-D-23-42998

Effectiveness of Deprescribing Interventions in Older Adults: An Overview of Systematic Reviews

The authors have synthesized evidence from reviews to provide an overarching overview of effectiveness of Deprescribing interventions in older adults. This is a valuable piece of work and appreciate the authors’ effort. Some of my comments are as follows:

1. Introduction – L83: The authors state that earlier work focused on reducing polypharmacy by addressing the appropriateness of medications and discontinuing those where harms outweigh benefits. As a reader, I would want to look into some of these studies. The authors can highlight these studies providing reference(s).

Response: The point that we were trying to make with these sentences is that medication discontinuation has been a focus of research for many years, and with the focus of deprescribing, the process is more systematic, and patient centered. We have made the following revision to make this point clearer (lines 81-85).

“For decades, research has focused on mitigating the harmful effects of polypharmacy by reducing the number of medications and discontinuing those where harms outweigh benefits. More recently, deprescribing has emerged as a systematic approach for improving the quality of medication use that is patient-centered and is aligned with the 4Ms of Age-Friendly care (mind, mobility, medications, and what matters most) [10].” 

2. Method – Search strategy - L120: The term information specialist might not be a common word in some parts of the world. It might be worth defining it or providing alternative word in a bracket such as ‘librarian’ for wider audience.

Response: We have clarified the term information specialist by adding the alternative term medical librarian in brackets (Line 118).

3. Method – Table 1 – Population: I appreciate that the authors have considered studies conducted in long-term care even if age was not provided. My comment is related to the definition of population. The authors have considered aged 60 years and above as older adults. However, the authors have used search term ‘Aged’ and ‘older adults’ in their search strategy, which are considered as electronic databases as person 65 years and above. Often studies on older adults have considered the later definition. So, it would be great if the authors could clarify this.

Response: We agree that usually the definition of older adults is 65 years and older. We had identified reviews with studies of lower mean age that were relevant. Given the lack of information for some older adult subgroups, we decided it was important to capture these. At times the use of older adult is used for studies that used a lower <65 cutoff. We have added this point as a potential limitation of the review in the Discussion (Line 467), as follows:

“We aimed to explore the effect of deprescribing in older adults and there was some variation on how this was reported so we used an age cut off point of 60 years to ensure we included as many reviews of older adults as possible.”

4. Method - Expert panel: The authors have considered an interprofessional expert panel for providing input in this review. I understand that they had a broad range of clinical and perspective related to the care of older adults. But for the methodological rigor the reader would benefit from knowing about the specific criteria for their inclusion.

Response: Thank you for your suggestion. We have revised this section with more details as requested (lines 189-196)

“We convened an expert panel of seven interprofessional members drawn from medicine, pharmacy, and nursing with specialization in geriatrics. We identified members that had clinical and/or research experience in optimizing and deprescribing medications in older adults. Panel members represent diverse practice settings (Department of Veterans Affairs, academic medical centers, ambulatory care, home visits) and specialties (oncology, internal medicine, geriatrics/palliative care). The panel advised on key steps (including the study design, interpretation and presentation of results and manuscript review). Potential conflicts of interest were assessed at the time of manuscript submission all members of the expert panel stated they had no conflicts of interest.

5. Result – Step 2 (page 12, L276): The author state there were 19 studies reporting on mortality but have cited only 18 studies.

Response: Thank you for catching this error. There were 19 studies and we have added the additional reference.

6. Discussion – please consider the grammar – L384: “…and those that did found…”

Response: Thank you for pointing this out. We have made the revision on line 391.

7. Discussion: The authors have raised a really good point that the majority of reviews did not specifically aim to examine harms. While this is true, studies reporting on the causality of harm due to deprescribing is also limited. It is also true that unless evidence of harm associated with deprescribing is well established, deprescribing option should be open to patients. This is not a comment but a view.

Response: Thanks for providing your view on this issue. We agree and have highlighted this in our Implications for Practice and Policy

Reviewer #3: Thank you for asking me to review this manuscript. This topic is important and timely. There are some comments below that I think should be seriously considered to ensure this overview meets current standards and provides more information to the reader.

Major:

a. Step 1: summarizing conclusions of reviews and meta-analyses: Although the authors have charted the review authors’ conclusions (in a supplement table), they do not perform any synthesis on the conclusions or even describe them at any length. I would suggest not consider

---

## [Editor Report · Decision Letter 1]

27 May 2024

Deprescribing Interventions in Older Adults: An Overview of Systematic Reviews

PONE-D-23-42998R1

Dear Dr. Gray,

We’re pleased to inform you that your manuscript has been judged scientifically suitable for publication and will be formally accepted for publication once it meets all outstanding technical requirements.

Kind regards,

Morteza Arab-Zozani, Ph. D.

Academic Editor

PLOS ONE

Additional Editor Comments (optional):

Thank you for your clarification